# The Immune Privilege of Cancer Stem Cells: A Key to Understanding Tumor Immune Escape and Therapy Failure

**DOI:** 10.3390/cells10092361

**Published:** 2021-09-08

**Authors:** Claudia Galassi, Martina Musella, Nicoletta Manduca, Ester Maccafeo, Antonella Sistigu

**Affiliations:** 1Dipartimento di Medicina e Chirurgia Traslazionale, Università Cattolica del Sacro Cuore, 00168 Rome, Italy; claudia.galassi@unicatt.it (C.G.); martina.musella@unicatt.it (M.M.); nicomandu123@gmail.com (N.M.); ester.maccafeo@unicatt.it (E.M.); 2Tumor Immunology and Immunotherapy Unit, IRCCS Regina Elena National Cancer Institute, 00144 Rome, Italy

**Keywords:** tumor microenvironment, cancer stem cells, immunogenicity, immune escape, immunotherapy

## Abstract

Cancer stem cells (CSCs) are broadly considered immature, multipotent, tumorigenic cells within the tumor mass, endowed with the ability to self-renew and escape immune control. All these features contribute to place CSCs at the pinnacle of tumor aggressiveness and (immune) therapy resistance. The immune privileged status of CSCs is induced and preserved by various mechanisms that directly affect them (e.g., the downregulation of the major histocompatibility complex class I) and indirectly are induced in the host immune cells (e.g., activation of immune suppressive cells). Therefore, deeper insights into the immuno-biology of CSCs are essential in our pursuit to find new therapeutic opportunities that eradicate cancer (stem) cells. Here, we review and discuss the ability of CSCs to evade the innate and adaptive immune system, as we offer a view of the immunotherapeutic strategies adopted to potentiate and address specific subsets of (engineered) immune cells against CSCs.

## 1. Introduction

Cancer is an evolving and dynamic disease whose progression may result in intra-tumoral heterogeneity (ITH) that confers different levels of sensitivity, and thus fuels resistance to conventional, targeted and immune-based therapies [1]. Cancer stem cells (CSCs, also known as tumor-initiating cells (TICs)), are widely considered relatively quiescent, immature, tumorigenic cells within the tumor mass, endowed with the capacity to self-renew and differentiate into multiple cell types (multipotency), and are responsible for ITH, tumor progression, recurrence, metastasis and therapy resistance [2,3].

CSCs were firstly identified in acute leukemia models as CD34^+^CD38^−^ cells [4,5]. Upon xenotransplantation into non-obese diabetic mice with severe combined immunodeficiency disease (NOD/SCID mice), CD34^+^CD38^−^ cells were shown to be significantly more tumorigenic than their differentiated counterparts [4,5]. Since then, a variety of studies proved the existence of cancer cells with stem-like properties in a broad spectrum of solid tumors [6].

The identification and isolation of CSCs is mainly based on the expression of surface proteins shared with normal stem cells (such as human embryonic stem cells (hESCs)), including the antigens CD34, CD38, prominin 1 (PROM1; best known as CD133), CD44, its variant isoform CD44v6, CD24, activated leukocyte cell adhesion molecule (ALCAM; best known as CD166), epithelial cell adhesion molecule (EPCAM), leucine rich repeat containing G protein-coupled receptor 5 (LGR5), ephrin receptors and the activity of the aldehyde dehydrogenases (ALDHs) [7,8] (Table 1). Despite these findings, a universal, highly sensitive and specific marker for CSCs is still lacking [8,9].

CSCs state and behavior are actively regulated by the aberrant activation of specific pathways, including Hedgehog (Hh), Notch and Wnt/β-catenin [10,11]. Additional pathways, such as the phosphoinositide 3-kinase/phosphatase and tensin homolog (PI3K/PTEN), control both CSC homeostasis and survival [12].

CSCs are naturally resistant to conventional chemotherapy and radiotherapy [13]. Moreover, CSCs are endowed with the ability to repair DNA damage through a robust DNA damage response (DDR) machinery [13] and to withstand high levels of replication stress by flaunting replication stress responses (RSRs) [14]. Another factor conferring therapeutic resistance to CSCs is the ability to enter dormancy [15] and acquire an anti-apoptotic state [16]. Based on these observations, many anti-cancer treatments (e.g., all-trans retinoic acid, lysine (K)-specific demethylase 1A (KDM1A) inhibitor, polycomb complex protein BMI-1 inhibitor, antibody targeting LGR5 or the delta-like canonical Notch ligand 3 (DLL3) and drug-like inhibitors of the Wnt/β-Catenin signaling) were approved to target and eradicate CSCs in tumors with different histological origins [17].

CSCs reside in protected niches, which components strongly influence their transcriptional and epigenetic signatures [18]. Moreover, CSCs are endowed with the ability to escape innate and adaptive immune control, a feature known as immune privilege [19], which will be extensively discussed in this review.

According to this, it was observed that genotoxic drugs, such as chemotherapy, induce cell death of differentiated cancer cells, while impacting on the surrounding tumor microenvironment (TME). The modifications inflicted in the TME affect cancer cell plasticity and may tip the balance between non-CSCs and CSCs in favor of stem cell fates [20,21], with significant effects on the clinical outcome [22].

CSC plasticity imposes the development of novel strategies aimed at impairing such properties. These specific interventions targeting the CSC-niche signals would prevent CSC (re)fueling and cancer elimination, while therapies exclusively based on intrinsic CSC features represent a short-term and temporary cancer (stem) cell eradication without the achievement of a long-lasting disease-free survival.

## 2. The Immune Privileged Status of CSCs

Since their discovery, the characterization of the immunological profile of CSCs was the subject of intensive studies, which showed that CSCs are immune privileged cells, and by virtue of this property, they play a key role in all the three phases of the immunoediting process [19]. Indeed, during the elimination phase, CSCs remain protected from the fully active and functional immune system. In the equilibrium phase, they enfold their low multiplication rate and high resistance to cell death/killing, while accumulating molecular alterations that favor the entry into the escape phase. During the escape phase, CSCs are allowed with the ability to proliferate, mostly through symmetric cell divisions, to recruit immunosuppressive cells, and to adopt other tricks to escape immune control [23,24]. In particular, enriched cells with stem-related markers isolated from patients with locally advanced head and neck squamous cell carcinoma (HNSCC) showed decreased expression of human leucocyte antigen I (HLA-I) molecule after chemotherapy treatment [25]. Accordingly, melanoma initiating cells displayed low levels of the major histocompatibility complex class I (MHC-I) [26]. Moreover, CSCs isolated from glioblastoma multiforme (GBM) were weakly positive and even negative for MHC-I [27], and TICs from the lung cancer cell line exhibited a lower expression of MHC-I as compared to differentiated counterparts [28]. The reduced expression of HLA-I or antigen processing machinery (APM) molecules confer cancer cells stem-like properties and protection from T cell recognition [29]. Although, opposing results were obtained from CSCs isolated from colorectal cancer (CRC) that were found to express HLA-I molecules [30].

The suboptimal expression of HLA-I molecules, if associated with detectable natural killer group 2D (NKG2D) ligands, can drive an increased susceptibility of CSCs to natural killer (NK) cells [31]. This phenomenon was observed in CSCs from CRC [32], melanoma [33], GBM [34] and oral squamous cell carcinoma [35]. However, CSCs from acute myeloid leukemia (AML) [36], GBM [27] and breast cancer (BC) [37] were shown to induce T cell polarization toward Th2 phenotype [27], and to escape NK cell-mediated killing by the downregulation of activating NKG2D ligands.

The upregulation of the “don’t eat me” signal CD47 is yet another trick that CSCs adopt to evade immune control [38]. CD47 is a broadly expressed transmembrane protein [39] that binds to the signal regulatory protein alpha (SIRPα) receptor on phagocytic cells (mainly macrophages (Mφs) and DCs), thus hindering phagocytosis [40]. Accordingly, the blocking of CD47 was shown to enable macrophage-mediated phagocytosis of CSCs from pancreatic ductal adenocarcinoma (PDAC) [41], AML [42] and hepatocellular carcinoma (HCC) [43], and thus promotes their elimination.

**Table 1 cells-10-02361-t001:** CSC markers in different cancers.

Marker	Tumor type	Rfs
**Surface markers**		
CD44 molecule	Lung, breast, gastric, liver, colorectal	[15,20]
Plasminogen sctivator, urokinase receptor (PLAUR, CD87)	Lung	[16]
Thy-1 cell surface antigen (THY1, CD90)	Lung, breast, gastric, liver	[17]
Prominin 1 (PROM1, CD133)	Lung, breast, gastric, liver, colorectal	[16]
Activated leukocyte cell adhesion molecule (ALCAM, CD166)	Lung, breast, gastric, liver, colorectal	[18]
Epithelial cell adhesion molecule (EpCAM)	Lung, colorectal	[19]
CD24 molecule (CD24)	Lung, breast, gastric, liver, colorectal	[20]
Integrin subunit beta 1 (ITGB1, CD29)	Breast	[21]
ITGA6 integrin subunit alpha 6 (ITGA6, CD49f)	Breast	[21]
Integrin subunit beta 3 (ITGB3, CD61)	Breast	[22]
CD70 molecule (CD70)	Breast	[23]
C-X-C motif chemokine receptor 4 (CXCR4)	Breast, gastric	[24]
Leucine rich repeat containing G protein-coupled receptor 5 (LGR5)	Breast, gastric, glioma, colorectal	[25]
Protein C receptor (PROCR)	Breast	[26]
Leucine rich repeat and Ig domain containing 2 (LINGO2)	Gastric	[27]
CD33 molecule (CD33)	AML, CML	[28,29]
Interleukin 3 receptor subunit alpha (IL3RA, CD123)	AML, CML	[28,29]
C-type lectin domain family 12 member A (CLEC12A, CCL1)	AML	[28]
HAVCR2 hepatitis A virus cellular receptor 2 (TIM3)	AML	[28]
Interleukin 2 receptor subunit alpha (IL2RA, CD25)	CML	[29]
Dipeptidyl peptidase 4 (DPP4, CD26)	CML	[29]
KIT proto-oncogene, receptor tyrosine kinase (KIT, CD117)	CML	[29]
CD36 molecule (CD36)	CML	[29]
Interleukin 1 receptor accessory protein (IL1RAP)	CML	[29]
**Intracellular markers and pathways**		
Aldehyde dehydrogenase (ALDH)	Lung, breast, gastric, colorectal, AML	[20]
Nanog homeobox (NANOG)	Lung, breast, gastric, liver, colorectal, AML	[33]
POU class 5 homeobox 1 (POU5F1, Oct-3/4)	Lung, breast, gastric, liver, colorectal, AML	[34]
SRY-box transcription factor 2 (SOX2)	Breast, gastric, liver, colorectal, AML	[35]
BMI proto-oncogene, polycomb ring finger (BMI1)	Breast	[36]
Wnt/β-Catenin signaling	Breast, liver, CML	[38]
JAK/STAT signaling	CML	[37]
FOXO signaling	CML	[29]
Hedgehog/Smo/Gli2 signaling	CML	[38]
Notch signaling	Breast, liver	[38]

Additional mechanisms of immune escape were identified and include: (i) the down-modulation of innate immune pathways (e.g., the toll-like receptor 4 (TLR4) [44], and the components of the signal transducer and activator of transcription 3 (STAT3) pathway [45]), and (ii) the heterogeneous expression of the immune checkpoints (e.g., Cytotoxic T-Lymphocyte Antigen 4 (CTLA4), Programmed death-ligand 1 (PD-L1), CD276 (also known as B7-H3) and V-set domain containing T cell activation inhibitor 1 (VTCN1; best known as B7-H4) [17]) (Figure 1).

Besides long-term self-renewal capability, multi-lineage differentiation, and high resistance to stress and apoptosis [46], CSCs are endowed with the ability to switch between dormant and proliferating states [46]. Based on their capacity to exit the cell cycle and remain in the G0 phase (quiescence), CSCs can evade host anti-tumor immunity by (i) preventing immune detection, (ii) preventing immune activation and (iii) activating immune suppression [47]. Dormant cancer (stem) cells from various cancer types were reported to downregulate both MHC-I complex [48] and UL16 binding protein (ULBP) ligands [49], which confer them the ability to evade T and NK cells, respectively. Furthermore, CSCs exert immunosuppressive functions including the expression of the immune checkpoints such as CD274 (best known as PD-L1) and CD80 (also known as B7.1) by preventing T cell activity and cancer dormancy [50]. In addition, dormant cancer (stem) cells evade NK and T cell-mediated apoptosis through the genetic inactivation of the oncosuppressor caspase 8 (CASP8) and the death receptor Fas cell surface death receptor (FAS) [51]. Similarly, dormant cancer cells escape T cell induced apoptosis by deregulating the suppressor of cytokine signaling 1 (SOCS1) cascade and overexpressing the pro-tumorigenic cytokine interleukin (IL)-3 [52].

The immune privilege of CSCs is strictly dependent on niches. CSC niches are specialized areas within the TME that preserve CSC state and plasticity and protect them from immune attack [23]. Indeed, CSC niches are made of non-cancerous stromal cells (i.e., cancer-associated fibroblasts (CAFs), endothelial cells, mesenchymal stem cells) and cancerous non-stem cells that by providing cues in the form of secreted factors (i.e., cytokines, extracellular vesicles and extracellular matrix (ECM) components) and complex interplays protect CSC exclusive abilities to self-renew, progress and disseminate in secondary sites [53]. Of note, the constant and evolving interplay with microenvironmental players, endows cancer (stem) cells with the ability to corrupt infiltrating immune cells, thus guaranteeing their survival and progression into overt disease [54]. Indeed, the bidirectional interaction of cancer cells with the microenvironment is of vital importance in developing immune escaping variants as a natural consequence of spatial and nutrient competition and evolutionary forces, which finally fuel tumor heterogeneity [55]. Therefore, more insight into these evolutionary dynamics, as well as into the ability of cancer cells to orchestrate immune evasion and immune suppression, is crucial for a better understanding of CSC (immune) biology and thus for the development of more effective therapies.

Nonetheless, despite their immune privileged phenotype, mainly due to defective MHC-I and -II expression [26], CSCs were found to express some tumor associated antigens (TAAs) such as carcinoembryonic antigen (CEA) [56], human telomerase reverse transcriptase (hTERT) [57], survivin [58] and centrosomal protein 55 (CEP55) [30], and some cancer testis (CT) antigens such as DnaJ homolog subfamily B member 8 (DNAJB8) [59], olfactory receptor 7C1 (OR7C1) [60], brother of the regulator of the imprinted site variant subfamily 6 (BORIS sf6) [61], mucin 1 (MUC1) [62], promelanosome protein (PMEL; also known as gp100) [63] and cancer/testis antigen 1B (CTAG1B, also known as NY-ESO1) [64] (Figure 2). TAAs and CT antigens specifically expressed on the CSC surface may represent promising targets for immunotherapy purposes [65]. Undoubtedly, the identification of tumor specific antigens (TSAs, also known as tumor neoantigens (TNAs)), derived from non-synonymous mutations, solely expressed on CSCs could represent a more specific and efficient immune target [51]. Although a lot of efforts are focused on this field by exploiting and integrating omic studies, a deep knowledge of the expression of TNAs on CSCs is still lacking. Notably, CSCs bearing a somatic mutation in the “driver” gene SMAD family member 4 (SMAD4), may elicit antigen-specific T cell responses directed to both stemness and bulk cancer cells [66]. The identification of new patient-specific (unique) tumor antigens could be of invaluable help for the development and the optimization of immunotherapeutic treatments.

## 3. Immunomodulatory Traits of CSCs and Their Immune Context

CSCs reside in cellular niches (i.e., anatomically distinct regions within the TME) that maintain the stemness properties of these cells, preserve their phenotypic plasticity, protect them from the immune system, and fuel their metastatic potential [53]. The formation of a unique (cellular) environment promoted by developing tumors is settled by acellular (i.e., ECM proteins elastin and collagen) and cellular (i.e., CAFs, monocytes, myeloid derived suppressor cells (MDSCs), tumor associated macrophages (TAMs), endothelial cells and T cells) components [67]. Each player exerts a peculiar function on its own but together co-operate and synergize to develop a complex network that (spatially) surrounds and protects the CSCs, as shown in Figure 3 and explained below.

DCs are critical sentinels of immunity and key effectors of self-tolerance [68]. The balance between immunogenic versus tolerogenic behavior, mainly relies on DC maturation and external signals within the local milieu [69]. Briefly, under steady-state, immature DCs express very low levels of costimulatory molecules and prevent T cell activation [70]. In the presence of inflammatory stimuli (e.g., pathogens, damaged cells), immature DCs that had taken up and processed pathogen- or tumor-associated antigens become fully mature and able to cross-present antigens on MHC-I molecules and overexpress costimulatory molecules thus inducing an antigen-specific T cell response [71]. Additionally, DCs are involved in the formation of anti-tumor T and B memory responses [72], and were shown to also play a tumor promoting role [73]. Indeed, it was observed that high production of the chemokine (C-X-C motif) ligand 1 (CXCL1) by cancer and stromal cells recruits immunosuppressive DCs which favor CSC survival and proliferation [74].

Mφs are tissue-resident or infiltrated immune cells critical for innate immunity [75]. Several evidences suggested that Mφs may establish a dual relationship with cancer cells depending on their polarization towards tumor-reactive (M1 phenotype) or tumor-promoting (M2 phenotype) states [76]. M2 polarization occurs in response to IL-4, IL-10, IL-13 or hypoxia. M2 TAMs produce arginase 1 (ARG1), IL-10, transforming growth factor-beta (TGF-β), vascular endothelial growth factor (VEGF), matrix metallopeptidase 9 (MMP9) and prostaglandin E2 (PGE2), thus subverting anti-tumor adaptive immunity and promoting tumor development and spreading [77]. The crosstalk between CSCs derived from HCC patients and TAMs is orchestrated by the secretion of IL-6 via STAT3, which promotes the expansion of CSCs both in in vitro and in vivo settings [78]. Along with similar lines, the IL-6-JAK1-STAT3 signal transduction pathway exerts a fundamental role in the phenotypic shift of non-stem into stem cells [79]. Moreover, STAT3 signaling is involved in cancer stem-like cell maintenance both in GBM [80] and BC [81]. Similarly, IL-6 secretion converts non-CSCs into CSCs in BC [82] and prostate cancer models [83]. Another factor involved in the promotion and stimulation of CSCs is represented by the pleiotrophin (PTN) that is released from TAMs and exerts its function through its receptor protein tyrosine phosphatase receptor type Z1 (PTPRZ1) [84]. As mentioned above, TAMs produce the multifunctional cytokine TGF-β, which drives a dedifferentiation process of CRC stem cells through the induction of twist family basic helix-loop-helix (bHLH) transcription factor 1 (TWIST1) [85]. These findings underscore the significance of TAMs as important components of the CSC niche, thus targeting TAMs by inhibiting their receptor such as the myeloid cell receptors colony-stimulating factor-1 receptor (CSF1R) or the chemokine (C-C motif) receptor 2 (CCR2), which decreases the number of TICs, improves chemotherapy outcome and elicits anti-tumor T cell responses [86].

MDSCs are the most prominent myeloid cell population infiltrating cancers and fueling tumor progression by modulating cancer cell survival, angiogenesis, invasion and metastasis [87]. Both human and mouse MDSCs are divided in two main subgroups, exhibiting different phenotypical and functional properties, namely, monocytic (M-MDSCs) and polymorphonuclear/granulocytic (PMN-MDSCs) MDSCs [88]. MDSC recruitment to the tumor is mediated by different chemokines, in particular C-C Motif Chemokine Ligand 2 (CCL2), CCL5 and CXCL5 [88]. The crosstalk between MDSCs and CSCs is regulated by key factors encompassing (i) PGE2 [89], (ii) STAT3 and NOTCH [90], and (iii) miRNA101 [91].

Tregs are a subset of CD4^+^ immune T cells characterized by the expression of the master transcription factor forkhead box P3 (FOXP3) [92]. In physiological conditions, Tregs guarantee tolerance to self-antigens and prevent/suppress autoimmune reactions [92]. In cancer, Tregs fuel tumorigenesis and tumor progression, by impairing host immune defenses [93]. Moreover, Tregs are involved in the promotion of cancer stemness as shown in CRC [94] and BC [95] cells, where they induce the expression of the reprogramming factor SRY-box transcription factor 2 (Sox2) via nuclear factor kappa B (NF-kB)-CCL1 signaling.

T helper 17 cells (Th17) are yet another specialized subset of CD4^+^ T cells characterized by the production of IL-17 [96]. Th17 are the key mediator of cancer development and display both tumor-promoting and tumor-suppressing activity [97]. It was reported that immune cell-derived IL-17 regulates stem cell features of pancreatic cancer cells by increasing the embryonic stem cell markers doublecortin-like kinase 1 (DCLK1) and ALDH 1 family member A1 (ALDH1A1) [98].

CD8^+^ T cells are a subtype of T cells and the main effectors of cell-mediated adaptive immune responses [99]. Interferon-gamma (IFN-γ), an effector cytokine produced by activated CD8^+^ T cells, was shown to induce NSCLC stem cells in a dose-dependent manner [100]. Furthermore, the interaction between CD8^+^ T cells and BC cells increases the number of BC stem cells in a cell-to-cell contact- or (at least) proximity-dependent manner [101].

Notably, CSCs actively induce immune subversion within the TME [102]. CSCs from glioma biopsies were displayed to release growth differentiation factor 15 (GDF15), which contributes to cancer cell proliferation and immune escape [103]. Similarly, CSCs from CRC, HNSCC and GBM were found to release immunosuppressive cytokines, encompassing IL-4, IL-8, granulocyte-CSF, MΦ inhibitory cytokine-1 (MIC-1) and TGF-β, that impair the cytotoxic T lymphocyte (CTL) function [29]. In addition, CSCs were shown to promote the polarization of Mφs toward a M2 phenotype by the production of TGF-β [104], granulocyte-macrophage colony-stimulating factor (GM-CSF) [105], macrophage colony-stimulating factor (M-CSF) [106], and via the cyclooxygenase (COX)-2/PGE2 pathway [107]. In turn, M2 TAMs support the expansion and drug resistance of CSCs by producing the cytokine IL-6 [108]. CSCs were shown to recruit TAMs at the tumor site through the release of soluble factors such as periostin (POSTN) in GBM [109] and the activation of specific pathways such as the Hippo one in liver cancer models [110].

CSCs also dialog with CD8^+^ T cells, MDSCs and Tregs and actively participate in the induction of an immunosuppressive milieu, which fuel their immune evasion and malignant potential [102]. Indeed, CSCs from HNSCC were shown to inhibit T cell, Treg and MDSC proliferation, contextually impairing the Th1 response and improving the Treg response [29]. Furthermore, brain TICs are endowed with the ability to impair T cell functionality by secreting the ECM protein tenascin-C (TNC) [111]. It was reported that the recruitment of Tregs into tumors occurs through the sensing of CCL5 released by CSCs. In turn, Tregs create an anti-inflammatory environment by releasing high levels of IL-10 [112]. CSCs from BC, CRC and HNSCC can evade immune surveillance by increasing the expression of the immune checkpoint ligand PD-L1, which binds to its receptor PD-1 expressed on T cell surfaces thus inducing their exhaustion [113]. Another immune “brake” is represented by T cell immunoglobulin mucin-3 (TIM-3), a specific surface molecule found on leukemic stem cells [114]. This receptor was described as responsible for T cell suppression and MDSC expansion [115]. Similarly, it was observed that B7-H4 promotes brain CSC tumorigenicity while negatively regulating T cell-mediated immunity [116]. Along with this, the interaction between GBM CSCs and Mφs enhances the expression of B7-H4 via the IL6/JAK/STAT3 pathway and thus promotes immune suppression [117]. The observations reported above show that multiple mechanisms and molecular pathways are either up regulated or aberrantly activated in CSCs, which result in immune privilege. Therefore, the blockade of these signaling through mono- or combination therapies should be considered in order to rescue the tumor-specific immune responses.

While experimentally defining CSC-immune cell interplays is undoubtedly challenging for a variety of reasons, including the intrinsic heterogeneity of CSC subsets, the difficulties related to in vitro co-culturing settings, efforts have been made to functionally characterize this crosstalk in multiple cancers and multiple settings (Box 1) and have revealed a series of cell-intrinsic and cell-extrinsic molecular mechanisms that underlie the CSC immune privilege and host co-evolution.

Box 12D and 3D models to evaluate the bidirectional crosstalk between CSCs and immune cell subsets.Conventional in vitro platforms, such as 2D monolayer cultures and transwell assays, have been widely utilized to understand the role of cancer stem cells (CSCs) within the tumor microenvironment (TME) and the mechanisms governing their immune suppression [118]. Over the past years, a flurry of studies in the field of “2D onco-immunology”, revealed the molecular pathways and (immune) effectors involved in the crosstalk between cancer cells with stem-like properties and different subsets of immune cells, including tumor associated macrophages (TAMs), dendritic cells (DCs), T regulatory cells (Tregs), αβ and γδ T cells [117]. Although the results collected from the aforementioned studies corroborate the immune privileged status of CSCs, 2D models are end-point assays that do not allow real-time observations of dynamic cellular interactions [119]. Most importantly, these models do not fully recapitulate the complex in vivo TME leading to non-physiologically relevant cell behavior [120]. Therefore, the innovations and development in 3D culture systems over the past 5 years allowed to better understand the complex interplay occurring between CSCs and immune effectors. However, a deep knowledge of the behavior of CSCs and their surrounding (immune) cells is still lacking. A practical example of a 3D culture is represented by human organ culture (HOC) consisting of a tissue that after a mechanical fragmentation process (<1 mm^3^ fragments) and resting for up to 24 h in a sterile 96-well plate containing culture inserts can be either immersed in fixative (paraformaldehyde for light microscopy or glutaraldehyde for electron microscopy) for in situ studies or frozen in isopentane-cooled in liquid nitrogen for molecular biological/biochemical analysis [121]. As compared to 3D cultures, HOCs emulate the natural biochemical and physical properties of the extracellular matrix (ECM) recognized as an independent factor that influences cell activity [122]. Furthermore, ECM scaffold of the former may contain biological pathogens and its vascular networks lacks segmental structure [123]. In comparison to organoids, HOCs preserve surrounding tissue that favors cell-to-cell and cell-to-ECM crosstalk and display histological diversity [124]. HOCs can be used to study (cancer) stem cells and their niche. It was reported that HOCs allowed the comparison of differentiated and kidney CSCs [125]. The latter showed low proliferation rate and may be hit by a cytotoxic drug coupled with a target agent (cyclophosphamide and the tumor necrosis factor receptor 2 signaling (R2TNF) agonist) able to induce cellular division, thus enhancing the efficiency of chemotherapy for renal clear cell carcinoma (RCC). Since HOCs offer the possibility to preserve the integrity of cells and matrices in an organ-specific structure reflecting better the in vivo complexity than 3D cultures and organoids, they may be exploited to deeper study the interaction of CSCs and different immune subsets.Another interesting 3D culture is represented by the hanging drop spheroid model that allows the formation of stable spheroids in a non-adherent 3D in vitro environment [126]. These spheroids were generated on a hanging drop array plate in which one or more cell populations were harvested and maintained in standard culture conditions for different studies including flow cytometry analysis, quantitative PCR (qPCR) and drug toxicity assays [127]. This 3D culture system was used to mimic the environment of ovarian cancer in which cancer cells interact with Mφs in anchorage independent conditions and grow as spheroids within the malignant ascites [128]. Furthermore, the hanging drop spheroid model was employed to investigate the interaction driven by the WNT pathway between ovarian CSCs and macrophages (Mφs) [129].Finally, an emerging 3D cell-culture model is represented by organ-on-a-chip, a “tissue chip” that mimics the microstructure, dynamic mechanical properties and biochemical functionalities of living organs [130]. Although this model seems to be an artificial model relying on microfabricated scaffolds to mimic ECM, it potentially allows to reproduce various tumor regions including the niche, which hosts cancer cells with stemness properties.

## 4. Immunotherapeutic Strategies against CSCs

The recognition of the yin and yang role of the immune system in both controlling and favoring tumor progression together with a growing knowledge of its functions, paved the way for the development of several immunotherapeutic strategies against cancer [131]. Immunotherapy could be basically defined as a refined therapeutic approach that employs the immune system’s power to specifically hit and (hopefully) eradicate cancer. A flurry of strategies empowering immune responses against cancer (stem) cells were investigated [132]. However, novel approaches aimed at arming T cells against specific CSC surface targets, which could help eradicate residual disease and potentially improve the long-term outcome of patients, are still in an embryonic phase and need further investigation and optimization. Here, we discuss some potential strategies to immunologically target CSCs.

Adoptive cell therapy (ACT) represents a (maybe the most) promising therapy for cancer patients. This clinical approach involves the ex vivo activation and expansion of autologous or allogeneic immune effector cells, followed by the (re)infusion into patients that were previously subjected to lymphodepleting chemotherapy or radiotherapy [133]. The T cell-based ACT can rely on CTLs, alone or combined with CD4^+^ helper T cells. Alternatively, ACT can involve NK or cytokine-induced killer (CIK) cells, both targeting cancer cells in a MHC-unrestricted manner [134]. Recently, the use of engineered T cells and chimeric antigen receptor (CAR) T cells to target EpCAM antigen showed therapeutic benefit in solid and hematologic tumor models [135]. Notably, this benefit was always associated with the depletion of CSCs [135]. Additionally, adoptively transferred CTLs specific for the CRC-stem cell antigen ankyrin repeat and SOCS Box containing 4 (ASB4) were reported to selectively kill CSCs [136]. The transmembrane glycoprotein CD133 is a common CSC marker and its expression on GBM stem cells has been exploited to develop an AC133-specific CAR cell able to kill CD133^+^ cells both in in vivo and in vitro settings [137]. Due to the therapeutic potential of CD133 for anti-cancer therapy, autologous CAR-modified T cells directed against CD133 were tested in a phase I clinical study [138]. Of note, 21 out of 23 metastatic patients who received CAR-T-133 cell-infusion had no developed detectable de novo lesions after the treatment. Similarly, CAR-T cells targeting the epidermal growth factor receptor variant III (EGFRvIII) were successfully tested against GBM stem cells [139]. Based on these data, a γ-retroviral vector expressing this EGFRvIII CAR was produced for clinical application (NCT01454596). Other CSC markers targeted by CAR-based ACT include the type II transmembrane receptor NKG2D, the interleukin three receptor alpha CD123 molecule and the disialoganglioside (GD2), all employed to develop CAR-T cells against both differentiated and CSCs, in GBM, AML and BC, respectively [140]. A combined CAR-T therapy co-targeting EGFR and CD133 in a case of unresectable/metastatic cholangiocarcinoma (CCA) resistant to chemo- and radiotherapy, was shown to induce a clinical response, although associated with several toxicities [141], which point the need to further investigate and ameliorate this biological therapy.

As mentioned above, ACT may be also based on the adoptive transfer of CIK cells [142]. CIK cells are non-MHC restricted, cytotoxic anti-tumoral cells expanded in vitro from T lymphocytes with the addition of IFN-γ and the monoclonal antibodies against CD3 and IL-2 [143]. CIK cells share characteristics of both T and NK cells, due to the expression of functional T cell receptor (TCR) and NK molecules [144]. It was reported the development of autologous CIK cells against BRAF inhibitor-surviving melanoma CSCs [145]. A cocktail of DCs and CIK cells was shown to inhibit the growth of hepatic [146] and prostatic [147] CSCs both in vitro and in vivo. In HCC and nasopharyngeal carcinoma (NPC), CIK cells were shown to eliminate CSCs via NKG2D-ligand recognition [148]. Accordingly, the adoptive transfer of heterologous NK cells showed the killing of both differentiated and undifferentiated cancer cells upon activation with IL-2 and IL-15 in various cancer models [149]. Additionally, a subset of sarcoma CSCs, that survived chemotherapy and molecular targeted therapy, was responsive to CIK immunotherapy [150]. However, as previously discussed in this review, CSCs may and do adopt strategies to escape immune control [151]. Indeed, epithelial ovarian CSCs were shown to evade CIK-mediated cellular lysis by activating hypoxia inducible factor-1α (HIF1A)-associated TGF-β1/decapentaplegic homologs (SMADs) and VEGFA signaling pathways that ultimately mediate the downregulation of intercellular adhesion molecule-1 (ICAM-1). As regards to immunotherapy-based combinations, in BC it was observed a synergistic effect between γδ and αβ CD8^+^ T cells occurring via the inhibition of farnesyl pyrophosphate synthase (FPPS), which allows it to overcome the resistance of CSC-like cells to γδ T cells. These cells, in turn, upregulate MHC class I and CD54 on CSC-like cells via secretion of IFN-γ and thereby increase the susceptibility to antigen-specific killing by αβ CD8^+^ T cells [152]. By exploiting the therapeutic potential of TCR-engineering, ALDH1A1-specific CD8^+^ T cells eliminate ALDH^bright^ cells and inhibit tumor growth and spreading in different murine cancer models [153]. Additionally, DNAJB8 (a cancer/testis antigen)-specific CTL clone efficiently recognizes CRC stem cells both in in vitro and in vivo settings [59].

Some of these studies have proved the anti-tumor effects of ACT by targeting CSCs, although overall studies in this area are rather limited. Importantly, a high rate of severe toxicities was observed for CAR-T therapies targeting TAAs including carbonic anhydrase IX (CA IX) in renal cell carcinoma (RCC) [154] and ERBB2 in metastatic CRC [155]. Indeed, while CAR-T cells are among the most promising anti-cancer therapies, their widespread clinical use is prevented by various limitations encompassing: (i) the “on-target on-tumor” toxicities, characterized by moderate-to-severe and even fatal forms of cytokine release syndromes and/or by excessive detrimental tumor cell necrosis, so-called tumor lysis syndrome, which can cause a broad spectrum of systemic metabolic disturbances [154]; (ii) the “on-target off-tumor” toxicities, due to a shared expression of targeted antigens by both cancer and normal cells; (iii) the “off-target” toxicities, triggered by the interaction between the extracellular crystallizable fragment (Fc) of CARs and the Fc receptor (FcR) expressed on innate immune cells, which leads to antigen-independent activation [156]; (iv) a suboptimal trafficking to tumors and in vivo persistence [157]. A plethora of promising approaches are currently under development to overcome CAR-T cell related toxicities and thus improve and broaden their application to solid tumors, as extensively reviewed in [158].

DC-based vaccine immunotherapy takes advantage of DCs, the most powerful APCs that generate robust immune responses, due to their capacity to link innate and adaptive immunity [159]. Different DC vaccine loading approaches have been tested in clinical trials including (i) loading of DCs with peptides, proteins, and tumor lysates; (ii) mRNA transfection; (iii) delivery of DNA and the use of viral vectors [160]. During the last decade, several clinical trials were developed including DCs loaded with tumor cell lysates in distinct tumor types such as metastatic hormone-refractory prostate cancer [161] and melanoma [162]. Furthermore, the use of DCs loaded with tumor-derived RNAs was shown highly immune-stimulating in patients with metastatic melanoma [163]. Accordingly, DCs pulsed with CSC lysates were reported to trigger potent anti-tumor immune responses, in malignant melanoma, renal, pancreatic and squamous cell carcinoma [159]. Moreover, DCs charged with total lysates of Panc-1 CSCs and loaded with Nanog homeobox (NANOG) peptide evoke strong anti-tumor responses in pancreatic and ovarian cancer, respectively [164]. Notably, ALDH^(high)^ CSC-DC vaccine was reported to reduce local tumor relapse and to increase host survival in squamous cell cancer and melanoma murine models [165]. The combination of DC based vaccines with immune checkpoint blockers (anti-CTLA-4 and anti-PD-1/PD-L1) showed an improved targeting and a successful eradication of CSCs in melanoma, GBM and bladder cancer models [159]. Therefore, the use of poly antigenic tumor lysates from CSCs would potentially allow the simultaneous targeting of the whole spectrum of antigens and would thus be a way to overcome resistance due to the loss of one or a few antigens.

Oncolytic virotherapy is based on natural or genetically engineered non-pathogen viral strains that directly kill tumor cells through immunogenic cell death (ICD) and thus switch on the so called “cancer-immunity cycle” [166]. Oncolytic virotherapy gained attention over the past decade owing to the ability of oncolytic viruses to interact with cancer cells and immune cells within TME (and, hence, keep clinical response long-lasting) and to the simply “editing” of the viral platform, which allows the implementation of various genetic elements encoding immunostimulatory cytokines (e.g., IL-7 and IL-12) [167]. The first oncolytic virus to obtain Food and Drug Administration (FDA) approval was talimogene laherparepvec to treat patients with advanced melanoma [168]. Although the precise mechanisms underlying oncolytic virus-mediated productive crosstalk between immune and cancer cells remain to be elucidated, improvements in developing immune cell-mediated anti-tumoral activity adopting oncolytic viruses are constantly being made. Genetically engineered oncolytic viruses are being developed and showed therapeutic efficacy as they specifically target CSCs in several solid tumor models [159]. Indeed, the use of engineered viruses to target such otherwise resistant cancer cells is a hot field of research.

Immunotherapy-based combinations targeting CSCs may exert greater therapeutic effects than monotherapies. Since immunotherapeutic regimens only benefit from a subset of patients, in the melanoma murine model it was investigated the efficacy of the ALDH CSC-DC vaccine combined with a dual blockade of PD-1 and CTLA-4 [169]. Mice who received the triple combination displayed a stronger tumor growth control and even more regression than those who received the CSC-DC vaccine alone. Furthermore, the combination of two monoclonal antibodies against CTLA-4 and PD-1 molecules with oncolytic virus expressing IL-12 induced the recruitment of Mφs into TME and their polarization toward an inflammatory phenotype, along with an increment of T effector cells [170]. In an orthotopic mouse model of glioma, the combinatorial administration of a vaccine (containing GBM stem cell lysate, DCs and TLR-9 agonist CpG motif-containing oligodeoxynucleotides (CpG ODNs)) and the anti-PD-L1 antibody confers a greater survival advantage and decrement of the Treg fraction as compared to the vaccine alone [171]. Similarly, anti-PD-1 potentiates the efficacy of a vaccine targeting bladder CSCs by boosting the recruitment of pro-inflammatory immune cells and thus improving host survival [172]. Similarly, oncolytic virotherapy combined with the targeting of the VEGF receptor (VEGFR) tyrosine kinase inhibitor (TKI) was shown to be more efficient than each monotherapy [173].

CSC resistance to the standard anti-tumor treatments dictates the need to generate novel targeted therapeutic interventions aimed at eradicating CSCs. The knowledge of CSC immunological properties, as well as their sensitivity to immunotherapeutic approaches, are somehow limited but promising immunological targets in pre-clinical and clinical settings are emerging. Therefore, this new knowledge may drive next-generation therapies and, more likely, immunotherapies able to target CSCs and thus render cancer cure a reality.

## 5. Concluding Remarks

Scientific research on CSCs has contributed to their genetic, cytogenetic and phenotypical characterization and has brought to light their immune-privileged nature that allows them to escape from destruction by the innate or adaptive immunity. The peculiar nature of CSCs (i.e., self-renew, multipotency and therapy resistance) favors the development of resistance to conventional therapies. In addition, CSCs displayed low immunogenicity (i.e., MHC-class I and APM downregulation, NK ligand and CD47 downregulation, dormancy, etc.), and the currently ongoing immunotherapeutic approaches fail to selectively target them. This bleak scenario could be improved by understanding the mechanisms regulating the genomics, epigenetics and immunology of CSCs in order to shape the immune system toward effective anti-tumor immune surveillance. These data could suggest the way to eradicate tumors by optimizing and developing novel immunologic approaches directed toward specific targets of CSCs.

## Figures and Tables

**Figure 1 cells-10-02361-f001:**
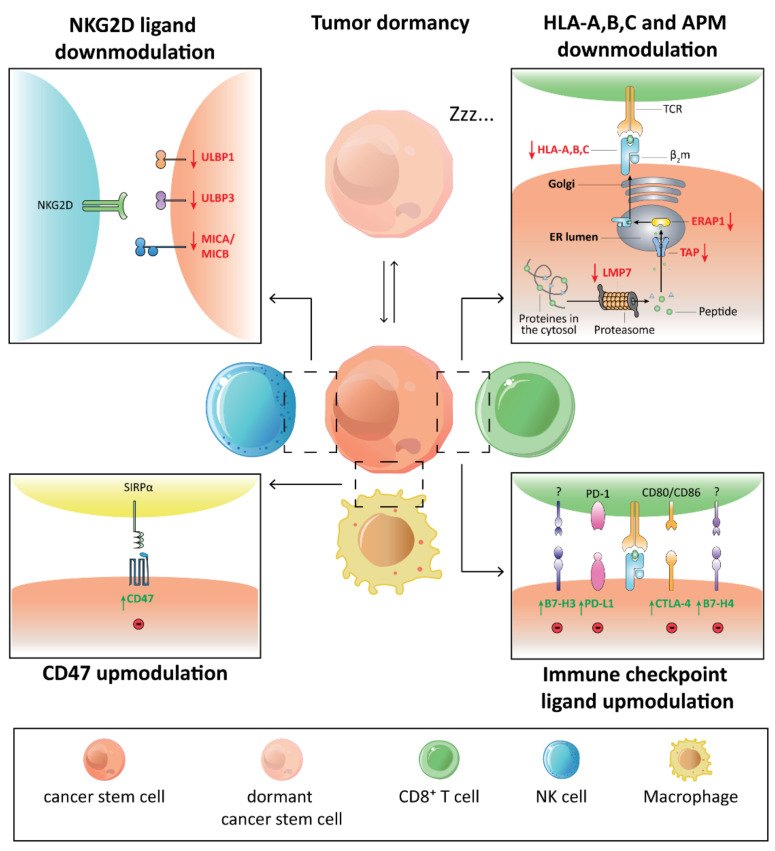
Immune escape mechanisms of CSCs. Cancer stem cells (CSCs) adopt several tricks to escape immune control. First, CSCs avoid the recognition by CD8^+^ T cells by reducing the human leukocyte antigen-A, B, C (HLA-A, B, C) and the antigen processing machinery (APM) molecule expression. Second, CSCs impair the cytotoxicity of natural killer (NK) cells by downregulating the natural killer group 2D (NKG2D) ligand expression. Third, the ligation of immune checkpoint ligands such as programmed death-ligand 1 (PD-L1), cytotoxic T-lymphocyte antigen-4 (CTLA-4), B7 homolog 3 (B7-H3) and B7 homolog 4 (B7-H4) expressed by CSCs to their respective receptors on T cells decreases their proliferation and interferon-γ (IFN-γ) production or leads to apoptosis. Fourth, CSCs upmodulate the “don’t eat me” signal CD47 that blocks their phagocytosis through the interaction with the membrane glycoprotein signal regulatory protein α (SIRP-α) expressed on the surface of macrophages (Mφs). Fifth, CSCs acquire a state of dormancy by entering into a G0 phase of cell cycle arrest. The cancer dormancy protects CSCs from immune control, ensuring their survival and eventually their metastatic dissemination.

**Figure 2 cells-10-02361-f002:**
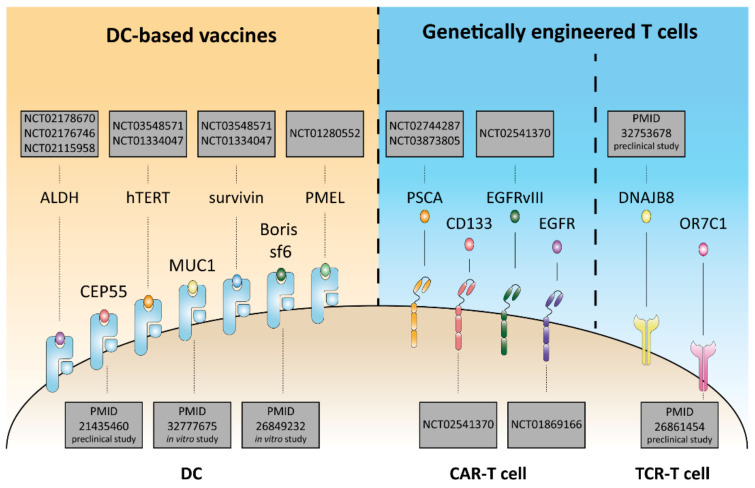
Immunotherapeutic targeting of CSC antigens. Cancer stem cells (CSCs) express several tumor-associated antigens (TAAs). CSCs are selectively targeted by means of dendritic cells (DCs) and genetically modified T cells (T cell receptor (TCR) T cells and chimeric antigen receptor (CAR) T cells). DC-based vaccines consist of DC expansion and loading with CSC specific antigen to prime CD8^+^ T cells, which, in turn, recognize and eliminate CSCs. TCR-engineered T cells are T cells equipped with a genetically modified TCR that specifically recognizes tumor antigen presented by human leukocyte antigen-A, B, C (HLA-A, B, C) on the surface of CSCs. CAR-T cells are T cells engineered to express a CD3ζ intracellular domain joined to an antibody receptor that recognizes tumor antigens exposed on CSC surface in an HLA-unrestricted manner. Some TAAs, such as aldehyde dehydrogenase (ALDH), human telomerase (hTERT) and survivin, and a cancer testis, (CT) antigen such as premelanosome protein (PMEL), are under clinical trial.

**Figure 3 cells-10-02361-f003:**
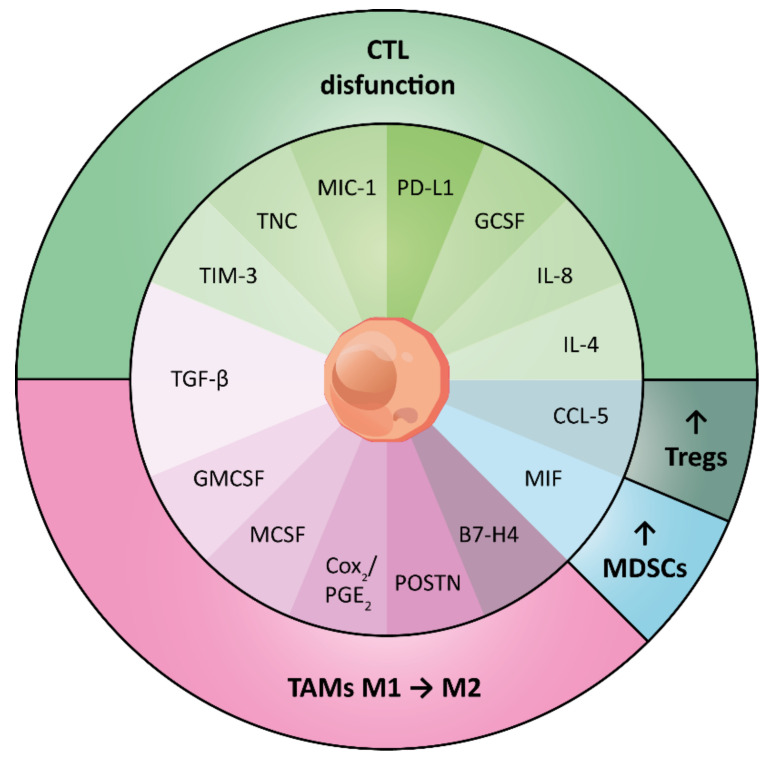
CSC immunomodulatory patterns impairing innate and adaptive immune system. A schematic model showing the plethora of cellular, molecular and physical factors either released from cancer stem cells (CSCs) within the tumor microenvironment (TME) or exposed by their surface that create an immune contexture crowded with immunosuppressive cells such as M2 tumor associated macrophages (TAMs), myeloid derived suppressor cells (MDSCs) and T regulatory cells (Tregs) and impaired cytotoxic T lymphocytes (CTLs).

## Data Availability

Not applicable.

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
