# Peer review of "The Immune Privilege of Cancer Stem Cells: A Key to Understanding Tumor Immune Escape and Therapy Failure"

_cells, 2021, doi:10.3390/cells10092361_

Round 1

Reviewer 1 Report

This article discusses the immune privilege of cancer stem cells as a potential key to understanding tumor immune escape and therapy failure. While the idea as such is not new, it is good to have an updated summary on this hypothesis, which is highly plausible, but not as abundantly discussed as many other theories that seek to explain immune-refractory properties of cancers. The article as such as well-written. Thus, I have just some minor criticisms:

1) The link between tumor immunoediting and the poor immunogenicity of CSC has initially been outlined in PMID: 25120546, which should be cited.

2) Tumor cell dedifferentiation is discussed. However, key publications on the topic are not cited: PMID: 23051752, PMID: 26530832, PMID: 16849459, PMID: 19276366, PMID: 30692216 should be cited.

3) Limitations of 2D systems and opportunities to overcome them should not be newly introduced in the conclusionsm, but discussed in an earlier paragraph.

4) The authors are too uncritical towards adoptive cell therapy. ACT can also prompt immune responses against non-cancer tissues, which has caused several fatalities with CAR T cells. Therefore, a more detailed assessment of speficity and of potential on- and off-target toxicities is needed.

5) With dendritic cells, the maturation status is crucial to achieve an immunogenic rather than tolerogenic antigen presentation. As the authors have chosen to provide a very basic introduction into the various immune cells, this should also be mentioned.

6) minor: line 313 "GMB" should probably be "GBM"

Author Response

Point-by-point reply to the Reviewer No. 1

We appreciate the reviewer’s efforts and the very detailed criticism that follows and to which we responded, point-by-point, as follows.

Minor point 1 raised by Reviewer No. 1: The link between tumor immunoediting and the poor immunogenicity of CSC has initially been outlined in PMID: 25120546, which should be cited.

Our response: Following the reviewer’s suggestion we cited the Manuscript by Valentin S Bruttel and Jörg Wischhusen.

Minor point 2 raised by Reviewer No. 1: Tumor cell dedifferentiation is discussed. However, key publications on the topic are not cited: PMID: 23051752, PMID: 26530832, PMID: 16849459, PMID: 19276366, PMID: 30692216 should be cited.

Our response: We cited the papers, as properly suggested.

Minor point 3 raised by Reviewer No. 1: Limitations of 2D systems and opportunities to overcome them should not be newly introduced in the conclusions, but discussed in an earlier paragraph.

Our response: The point is very well taken and the referee is right to address this point. We moved Box 1 entitled “2D and 3D models to evaluate the bidirectional crosstalk between CSCs and immune cell subsets” reporting current in vitro models and protocols to study the relationship between cancer and immune cells and the underlying pathways at the end of the chapter describing the “immunomodulatory traits of CSCs and their immune context”. We believed that a dedicated chapter could be out of the focus of the review, so that we opted for a supporting Box.

Minor point 4 raised by Reviewer No. 1: The authors are too uncritical towards adoptive cell therapy. ACT can also prompt immune responses against non-cancer tissues, which has caused several fatalities with CAR T cells. Therefore, a more detailed assessment of speficity and of potential on- and off-target toxicities is needed.

Our response: We thank the referee’s constructive criticism and address his/her point.

Minor point 5 raised by Reviewer No. 1:With dendritic cells, the maturation status is crucial to achieve an immunogenic rather than tolerogenic antigen presentation. As the authors have chosen to provide a very basic introduction into the various immune cells, this should also be mentioned.

Our response: We thank the referee’s constructive criticism and address his/her point.

Minor point 6 raised by Reviewer No. 1: minor: line 313 "GMB" should probably be "GBM"

Our response: We corrected the sentence, as properly suggested.

Reviewer 2 Report

This review on cancer stem cell (CSCs) is focused on their immune privilege status and interrogates ways of escape CSCs employ to evade the host immune system. The review begins with a description of known phenotypic and functional characteristics of CSCs in various cancers; it goes on to listing escape pathways potentially used for CSC escape from immune cells; proceeds to describing immune suppressive attributes of CSCs; and ends with consideration of immunotherapeutic strategies available today for silencing or eliminating CSCs.

 As such, the review provides an in-depth view of CSCs characteristics that allow them to survive and thrive in the TME. However, this is only one side of the coin involving the immune system. For CSCs to become self-renewing, multipotent, therapy resistant, non-immunogenic entities, they must be an integral part of a program that predicates tumor growth progression and metastasis. While undoubtedly the immune system plays a key role in modulating tumor progression, it is the inexorable immune suppression orchestrated by the tumor contra immune cells that enables it to win the battle. The authors point out that CSCs participate in inducing immune suppression that pervades the TME. The authors also acknowledge that we know relatively little about the CSC crosstalk with non-immune cells in the TME, including tumor cells. Yet, it is likely that it is the tumor, not the immune system, that determines CSC emergence as well as genotype and phenotype they acquire in the TME.

 Thus, while it is rational to talk and write about how CSC escape from the immune system, it is critically important to remind the reader that it is the tumor and not the immune system that orchestrates this escape. So, the emphasis should likely be on the tumor-CSCs crosstalk, which is the major “sculpturing” factor in the TME that immune cells have to deal with. As an example, the dormancy of CSCs is their well- known feature, but how exactly the dormancy is evoked in tumor cells which are transformed into CSCs remains unclear. Could the authors speculate as to potential mechanisms that mediate the tumor to CSC crosstalk as well as immune cells to CSCs crosstalk?  Increasingly, the role of extracellular vesicles (EVs) in the TME has been proposed as a mechanism of information transfer between various components in the TME. While still under investigation, EV driven reprogramming in the TME might provide the mechanistic underpinning for immune privilege of CSC.

The review is informative and is clearly written. The illustrations are also adequate. The list of references with over 250 entries is excessive.

Author Response

Point-by-point reply to the Reviewer No. 2

We thank the reviewer for the encouraging words. Specific comments have been addressed, as listed below.

Minor point 1 raised by Reviewer No. 2: … while it is rational to talk and write about how CSC escape from the immune system, it is critically important to remind the reader that it is the tumor and not the immune system that orchestrates this escape. So, the emphasis should likely be on the tumor-CSCs crosstalk, which is the major “sculpturing” factor in the TME that immune cells have to deal with… the role of extracellular vesicles (EVs) in the TME has been proposed as a mechanism of information transfer between various components in the TME. While still under investigation, EV driven reprogramming in the TME might provide the mechanistic underpinning for immune privilege of CSC.

Our response: We thank the reviewer to raise this point. Intrigued by his/her suggestion, we wrote a part within the chapter entitled “the immune privileged status of CSCs” briefly describing the intricate and dynamic interplays between cancer cells, CSCs and other components of the CSC niches within the TME which play critical roles in modulating immune recognition/evasion, as we cited prominent literature in this field.

Minor point 2 raised by Reviewer No. 2: The list of references with over 250 entries is excessive.

Our response: We thank the referee’s constructive criticism and address his/her point